# Federated Learning with Uncertainty and Personalization via Efficient Second-order Optimization

**Shivam Pal**                                                                 *pshivam@cse.iitk.ac.in*
*Dept. of Computer Science and Engineering*
*IIT Kanpur*

**Aishwarya Gupta**                                                     *aishwaryag@cse.iitk.ac.in*
*Dept. of Computer Science and Engineering*
*IIT Kanpur*

**Saqib Sarwar**                                                               *saqib@cse.iitk.ac.in*
*Dept. of Computer Science and Engineering*
*IIT Kanpur*

**Piyush Rai**                                                                 *piyush@cse.iitk.ac.in*
*Dept. of Computer Science and Engineering and Dept. of Intelligent Systems*
*IIT Kanpur*

**Reviewed on OpenReview:** *https: // openreview. net/ forum? id= TzhCnGBK4F*

## Abstract

Federated Learning (FL) has emerged as a promising method to collaboratively learn from decentralized and heterogeneous data available at different clients without the requirement of data ever leaving the clients. Recent works on FL have advocated taking a Bayesian approach to FL as it offers a principled way to account for the model and predictive uncertainty by learning a posterior distribution for the client and/or server models. Moreover, Bayesian FL also naturally enables personalization in FL to handle data heterogeneity across the different clients by having each client learn its own distinct personalized model. In particular, the hierarchical Bayesian approach enables all the clients to learn their personalized models while also taking into account the commonalities via a prior distribution provided by the server. However, despite their promise, Bayesian approaches for FL can be computationally expensive and can have high communication costs as well because of the requirement of computing and sending the posterior distributions. We present a novel Bayesian FL method using an efficient second-order optimization approach, with a computational cost that is similar to first-order optimization methods like Adam, but also provides the various benefits of the Bayesian approach for FL (e.g., uncertainty, personalization), while also being significantly more efficient and accurate than SOTA Bayesian FL methods (both for standard as well as personalized FL settings). Our method achieves improved predictive accuracies as well as better uncertainty estimates as compared to the baselines which include both optimization based as well as Bayesian FL methods.

## 1 Introduction

Federated Learning (FL) (McMahan et al., 2017) aims at learning a global model collaboratively across clients without compromising their privacy. It involves multiple client-server communication rounds, where in each round the selected clients send their local models (trained on their private dataset) to the server and the server aggregates the received models followed by its broadcasting to all clients. Thus, the global model, an approximation to the model obtained if all the data was accessible, depends significantly both on

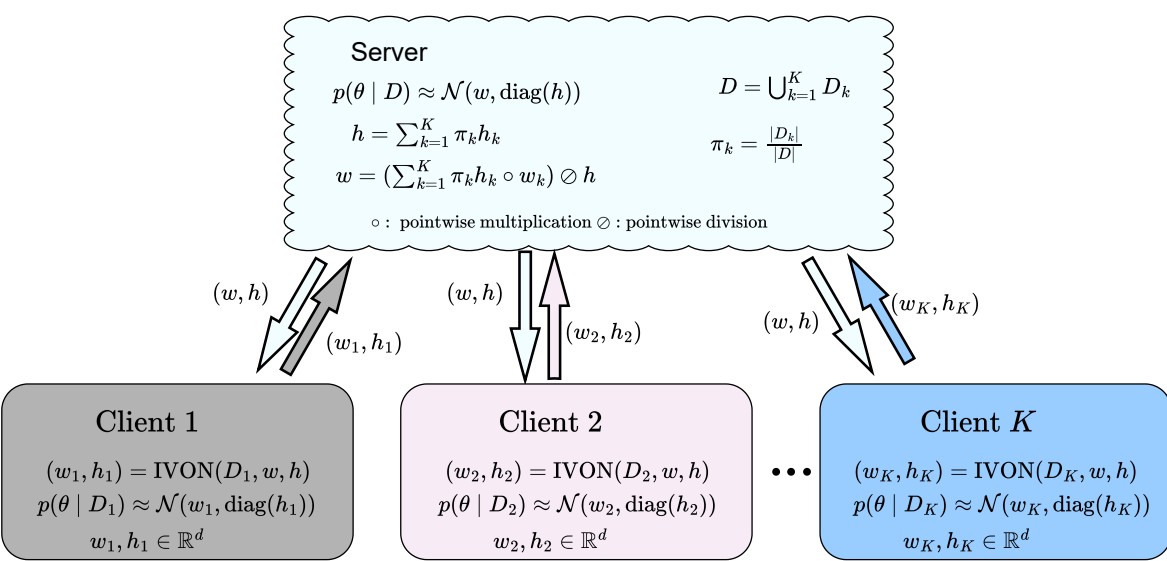

Figure 1: Illustration of FedIvon.

the quality of the received clients' models and the chosen aggregation strategy at the server. As a result, a straightforward approach like FedAvg (McMahan et al., 2017) can yield a high-performing global model if the data is i.i.d. distributed among clients but performs suboptimally in case of non-i.i.d. data distribution. Moreover, the challenges are compounded if the individual clients' private dataset sizes are small.

The limitations of standard FL become even more apparent with data heterogeneity, where clients have distinct data distributions. A single global model might fail to represent all clients well, leading to poor performance. This motivates personalized FL (pFL) Tan et al. (2022), which aims to adapt models to individual clients while leveraging shared global knowledge.

In such settings, learning the posterior *distribution* instead of a point estimate at each client results in enhanced performance and uncertainty measures, as demonstrated in several recent works, such as (Al-Shedivat et al., 2020; Liu et al., 2024; Bhatt et al., 2023; Guo et al., 2023) which have advocated taking a Bayesian approach to FL. Moreover, Bayesian FL is also natural for personalization because the server model can serve as a prior distribution in a hierarchical Bayesian framework, enabling easy personalization of client models using their respective client-specific likelihoods. However, existing Bayesian FL and pFL methods usually rely on running computationally expensive routines on the clients (e.g., requiring expensive MCMC sampling (Al-Shedivat et al., 2020), expensive Laplace's approximation which requires Hessian computations and inversions (Liu et al., 2024) on the clients, or methods based on learning deep ensembles (Linsner et al., 2021)), as well as expensive client-server communication (Kassab & Simeone, 2022) and aggregation at the server (note that, unlike standard FL, Bayesian FL would require sending the whole client posterior to the server). Due to such computational bottlenecks and communication overhead, Bayesian approaches lack scalability, especially for clients with limited resources and bandwidth.

Thus, to bridge this gap, we propose a novel Bayesian FL algorithm FedIvon (with its high-level idea illustrated in Fig. 1), that balances the benefits of Bayesian inference, such as enhanced performance, and quantification of predictive uncertainty, with minimal increase in computational and communication overhead. In particular, we leverage the IVON (Improved Variational Online Newton) algorithm (Shen et al., 2024) to perform highly efficient variational inference (VI) on each client by approximating its local posterior using a Gaussian with diagonal covariance. It uses the natural gradient to capture the geometry of the loss function for faster convergence. Moreover, it computes the Hessian implicitly, making our method computationally cheaper than other existing Bayesian FL and pFL methods that use explicit Hessian computation, e.g., Laplace's approximation (Liu et al., 2024), expensive MCMC sampling (Al-Shedivat et al., 2020; Bhatt et al., 2023), or even VI (Kassab & Simeone, 2022) at the clients. These local posteriors can be efficiently sent

to the server and the global posterior can be computed for which we also present local posterior aggregation strategies. Our main contributions are:

- We introduce an efficient Bayesian FL method **FedIvon** that uses an efficient second-order optimization approach for variational inference whose per-round compute matches first-order optimizers (e.g., Adam) while delivering higher predictive accuracy and better uncertainty estimation than state-of-the-art Bayesian and non-Bayesian FL methods.

- Our method also supports client-level model personalization naturally by leveraging a hierarchical Bayesian framework. Clients can use the server's posterior as prior to learn their private models, effectively balancing local adaptation with global knowledge sharing.

## 2 Related Work

FedAvg (McMahan et al., 2017), the foundational federated learning algorithm, approximates the global model as the weighted aggregation of locally trained client models, performing effectively with i.i.d. data distributions. Since then, numerous sophisticated and efficient algorithms have been proposed to handle more realistic challenges such as non-i.i.d. data distribution, heterogeneous and resource-constrained clients, and multi-modal data as explored in recent works (Lin et al., 2020; Pfeiffer et al., 2023; Zhang et al., 2023; Che et al., 2023; Liu et al., 2023). However, here, we will primarily restrict our discussion to Bayesian FL and personalized FL algorithms as they are the most relevant to our work.

**Bayesian Federated Learning** In federated learning, a key limitation of conventional point-estimate-based approaches is their susceptibility to overfitting in limited data settings and lack of predictive uncertainty estimates. To address this, Bayesian approaches have been advocated for federated learning, which involves the computation of clients' local posterior distribution followed by their aggregation at the server to compute the global posterior distribution, offering enhanced performance and quantification of predictive uncertainty. Unfortunately computing full posterior distribution on resource-constrained clients can be intractable, and also poses communication overheads due to the need of communicating the full posterior to the server. FedBE (Chen & Chao, 2020) mitigates the communication overhead by leveraging SWAG (Maddox et al., 2019) to learn each client's posterior but communicating only its mean. The server then fits a Gaussian/Dirichlet distribution to the clients' posterior mean and distills it into a single model to be communicated in the next round. However, the server-side aggregation in FedBE does not incorporate the client model uncertainty since the clients do not transmit the respective posteriors to the server. FedPA (Al-Shedivat et al., 2020) addresses this by learning a Gaussian distribution for each client and computing the mean of the global posterior at the server via posterior averaging. However, it ultimately discards the global covariance as the server broadcasts only a point estimate in subsequent rounds, so the aggregated posterior is not reused as a prior on the clients. In contrast, our method FedIvon maintain a diagonal Gaussian posterior at the server and explicitly sends both mean and (diagonal) precision back to the clients, which is then used as the prior at the client for the next round. This preserves a fully hierarchical Bayesian update across communication rounds. Similarly, FedLaplace (Liu et al., 2024) approximates each client's posterior as a Gaussian distribution, modeling the global posterior as a mixture of Gaussian, though further approximated by a single Gaussian by minimizing KL divergence. Moreover, dependence on Laplace's approximation makes FedLaplace heavy in terms of computation and communication costs.

**Second-order Optimization for Federated Learning** shows promise for improving convergence but is often limited by efficiency and communication overhead. Methods such as FedNL (Safaryan et al., 2021), which use privacy-preserving Hessian learning and compression, and second-order approaches incorporating global line search (Bischoff et al., 2021b), offer potential solutions to these challenges.

**Personalized Federated Learning** In the case of non-iid data distribution among clients, a single global model represents the average data distribution and diverges substantially from each client's local distribution. Consequently, the global model, though benefitted from collaborative learning, performs suboptimally for individual clients. Personalized federated learning addresses this by adapting a part or the whole model to the local data distribution explicitly. A typical approach is to split the model into two parts, a base model for global representation learning and a head model for personalized learning. FedPer (Achituve et al., 2021)

and FedRep (Collins et al., 2021) use this strategy, applying FedAvg for collaborative learning of the base model leveraged by the head for local data adaptation. Similarly, FedLG (Liang et al., 2020) splits the model into local and global components to learn local and shared representations, respectively. It shares the global parameters with the server while enhancing local parameters further using the unsupervised or self-supervised approach. PerFedAvg (Fallah et al., 2020) applies a Model-Agnostic Meta-Learning (MAML) (Finn et al., 2017) inspired framework to learn a shared model for faster adaptation to the client's data. pFedME (T Dinh et al., 2020) decouples personalized adaptation from shared learning by regularizing each client's loss function using Moreau envelopes. pFedBayes (Zhang et al., 2022) is a Bayesian approach that aims at learning the personalized posterior distribution of each client. In each round, pFedBayes computes client's posterior using the global model as the prior and sends it to the server for updating the global model. pFedVEM (Zhu et al., 2023) also computes the client's posterior by restricting it to the Gaussian family. However, it leverages the collaborative knowledge of other clients by assuming conditional independence among clients' models given the global model.

## 3 Bayesian FL via Improved Variational Online Newton

The standard formulation of FL is similar to distributed optimization except some additional constraints, such as no data-sharing among clients and server and a limited communication budget. Assuming $K$ clients, let $\mathcal{D} = \bigcup_{k \in [K]} \mathcal{D}_k$ be the total available data where $\mathcal{D}_k$ denotes the private data of client $k$. The objective of standard FL is to solve $\boldsymbol{\theta}^* = \arg\min_{\boldsymbol{\theta}} \sum_{k \in [K]} - \log p(\mathcal{D}_k \mid \boldsymbol{\theta})$. However, this optimization problem is not trivial as it requires access to each client's data which is not permitted in the federated setting. Thus, a multi-round approach is usually taken where clients learn their local models, send these local models to a central server which aggregates them into a global model, and send the global model to the clients to continue the next round of learning.

Unlike standard FL, which only learns a point estimate of $\boldsymbol{\theta}$, an alternative is to take a Bayesian approach and learn the *distribution* of $\boldsymbol{\theta}$ conditioned on data. The posterior distribution of $\boldsymbol{\theta}$ can be written as

$$p(\boldsymbol{\theta} \mid \mathcal{D}) \propto p(\boldsymbol{\theta}) \prod_{k \in [K]} p(\mathcal{D}_k \mid \boldsymbol{\theta}) \tag{1}$$

where $p(\boldsymbol{\theta})$ is prior distribution on $\boldsymbol{\theta}$ and $p(\mathcal{D}_k \mid \boldsymbol{\theta})$ is the likelihood w.r.t. client $k$. Assuming uniform prior $p(\boldsymbol{\theta})$, it can be trivially shown that optimizing the standard FL objective function is equivalent to finding the mode of the posterior $p(\boldsymbol{\theta} \mid \mathcal{D})$, i.e., $\boldsymbol{\theta}^* = \arg\max_{\boldsymbol{\theta}} \log p(\boldsymbol{\theta} \mid \mathcal{D})$.

Computing the full posterior $p(\boldsymbol{\theta} \mid \mathcal{D})$ is more useful than computing just the point estimate $\boldsymbol{\theta}^*$ because the posterior helps take into account model uncertainty. However, it is computationally intractable to compute the posterior exactly. Directly approximating $p(\boldsymbol{\theta} \mid \mathcal{D})$ using approximate inference methods such as MCMC or variational inference (Angelino et al., 2016) is also non-trivial, as it requires computing each client's likelihood which in turn requires global access to all the client's data.

**Claim 1** *Given a common prior and conditional independence, the global posterior $p(\boldsymbol{\theta} \mid \mathcal{D})$ can be approximated at the server by the product of local client posteriors without requiring access to any client's local data.*

Proof of the claim 1 is given in the appendix B. If local posteriors $p(\theta \mid \mathcal{D}_k)$ are also being approximated, multiple rounds of optimization are needed to reduce the aggregation error in the global posterior (Al-Shedivat et al., 2020). In Bayesian FL, another challenge is to make the computation of the local posteriors, their aggregation at the server, and the client-server communication, efficient, which in general can be difficult in general (Al-Shedivat et al., 2020).

## 4 Client's posterior approximation

Assuming client $k$ has $N_k$ training examples, its local loss can be defined as $\bar{\ell}_k(\boldsymbol{\theta}) = \frac{1}{N_k} \sum_{i=1}^{N_k} \ell_i(\boldsymbol{\theta})$, and we can compute the point estimate of the parameters as $\boldsymbol{\theta}_k^* = \arg\min_{\boldsymbol{\theta}} \bar{\ell}_k(\boldsymbol{\theta})$. However, in our Bayesian FL

setting, we will compute the (approximate) posterior distribution for each client using variational inference, which amounts to solving the following optimization problem

$$q_k^*(\boldsymbol{\theta}) = \underset{q_k(\boldsymbol{\theta})}{\arg\min}\, \mathcal{L}_k(q) \tag{2}$$

$$\mathcal{L}_k(q) = \mathbb{E}_{q_k(\boldsymbol{\theta})}[\bar{\ell}_k(\boldsymbol{\theta})] + \mathbb{D}_{KL}(q_k(\boldsymbol{\theta}) \| p_k(\boldsymbol{\theta})) \tag{3}$$

where $p_k(\boldsymbol{\theta})$ is the prior and $\mathbb{D}_{KL}$ is the Kullback-Leibler divergence. If we use the Gaussian variational family for $q_k(\boldsymbol{\theta})$ with diagonal covariance then $q_k(\boldsymbol{\theta}) = \mathcal{N}(\boldsymbol{\theta}|\boldsymbol{m}_k, \mathrm{diag}(\boldsymbol{\sigma}_k^2))$, where $\boldsymbol{m}_k$ and $\boldsymbol{\sigma}_k^2$ denote the variational parameters that are to be optimized for. Optimizing the objective in Equation 3 w.r.t these variational parameters requires making the following updates

$$\boldsymbol{m}_k^{t+1} = \boldsymbol{m}_k^t - \alpha \hat{\nabla}_{\boldsymbol{m}_k} \mathcal{L}_k(q) \tag{4}$$

$$\boldsymbol{\sigma}_k^{t+1} = \boldsymbol{\sigma}_k^t - \alpha \hat{\nabla}_{\boldsymbol{\sigma}_k} \mathcal{L}_k(q) \tag{5}$$

where $\alpha > 0$ is the learning rate.

Computing exact gradients in the above update equations is difficult due to the expectation term in $\mathcal{L}_k(q)$. A naïve way to optimize is to use stochastic gradient estimators. However, these approaches are not very scalable due to the high variance in the gradient estimates. Shen et al. (2024) improved these update equations and provided much more efficient update equations similar to the Adam optimizer, which is essentially the improved variational online Newton (IVON) algorithm, with almost similar computational cost as Adam, and their key differences are summarized below.

- Unlike Adam which solves for $\boldsymbol{\theta}$, IVON solves for both the mean vector $\boldsymbol{m}$ and the variances $\boldsymbol{\sigma}^2$ which provides us an estimate of the Gaussian variational approximation at each client. Note that the mean $\boldsymbol{m}$ plays the role of $\boldsymbol{\theta}$ in Adam. In addition, the variances naturally provide the uncertainty estimates for $\boldsymbol{\theta}$, essential for Bayesian FL (both in estimating the client models' uncertainties as well as during the aggregation of client models at the server).

- Unlike Adam which uses squared minibatch gradients to adjust the learning rates in different dimensions, IVON uses a reparametrization defined as gradient element-wise multiplied by $(\boldsymbol{\theta} - \mathbf{m})/\boldsymbol{\sigma}^2$ to get an unbiased estimate of the (diagonal) Hessian. Using this, IVON is able to get a cheap estimate of the Hessian, which makes it a second-order method, unlike Adam.

- IVON offers the significant advantage of providing an estimate of second-order information $h$ with minimal computational overhead. The Hessian $\boldsymbol{h}$ corresponds to the inverse of $\boldsymbol{\sigma}^2$, where $\boldsymbol{\sigma}^2 \propto \frac{1}{\boldsymbol{h}}$. An estimate of $\boldsymbol{h}$ is accessible throughout the training process (see Algorithm 1). Moreover, there is no explicit update equation for $\boldsymbol{h}$. It is computed implicitly using gradient information. In comparison, standard optimization methods such as SGD, Adam, and SWAG require additional effort to estimate second-order information.

## 5 Posterior aggregation at server

At the server, we can aggregate the client posteriors to compute the global posterior (Fischer et al., 2024). IVON approximates clients' posteriors as Gaussians and product of Gaussian distributions is still a Gaussian distribution up to a multiplicative constant. Thus we approximate the global distribution as a Gaussian whose optimal mean and covariance matrix expressions are given below. Moreover, since each client's variational approximation is a Gaussian with diagonal covariance matrix, it makes the aggregation operations efficient. Let's assume $q(\boldsymbol{\theta} \mid \mathcal{D}_k) = \mathcal{N}(\boldsymbol{\theta} \mid \boldsymbol{\mu}_k, \boldsymbol{H}_k^{-1})$ where $\boldsymbol{\mu}_k = \mathbf{m}_k$ and $\boldsymbol{H}_k = \mathrm{diag}(\boldsymbol{h}_k) = \mathrm{diag}(1/\boldsymbol{\sigma}_k^2)$. Using results of the product of Gaussians (Liu et al., 2024; Fischer et al., 2024), our aggregation is defined as

$$\log q(\boldsymbol{\theta} \mid \mathcal{D}) \approx \sum_{k=1}^{K} w_k \log q(\boldsymbol{\theta} \mid \mathcal{D}_k) \tag{6}$$

where $w_k = \frac{N_k}{\sum_{k=1}^K N_k}$, $q(\boldsymbol{\theta} \mid \mathcal{D}) \approx \mathcal{N}(\boldsymbol{\theta} \mid \boldsymbol{\mu}, \boldsymbol{H}^{-1})$, with $\boldsymbol{H} = \sum_{k=1}^K w_k \boldsymbol{H}_k$ and $\boldsymbol{\mu} = \boldsymbol{H}^{-1} \sum_{k=1}^K w_k \boldsymbol{H}_k \boldsymbol{\mu}_k$.

---

**Algorithm 1** Client_Update

---

**Input:** Local dataset $D$, model weights $\boldsymbol{m}$, Hessian($\boldsymbol{h}$), local_epochs($E$), learning rates $\{\alpha_e\}$, weight decay $\delta$, hyperparameters $\beta_1, \beta_2$, batch-size $B$

**Output:** Trained model weights $\boldsymbol{m}$, Hessian $\boldsymbol{\sigma}$

1: $\mathbf{g} \leftarrow 0, \quad \lambda \leftarrow |D|, \quad n = E * |D|/B$.
2: $\boldsymbol{\sigma} \leftarrow 1/\sqrt{\lambda(\mathbf{h} + \delta)}$.
3: $\alpha_e \leftarrow (h + \delta)\,\alpha_e$ for all $e \in \{1, 2, \ldots, n\}$.
4: **for** $e = 1$ to $E$ **do**
5:      Sample a batch of inputs of size $B$ from $D$.
6:      $\widehat{\mathrm{g}} \leftarrow \widehat{\nabla}\,\ell(\boldsymbol{\theta})$, where $\boldsymbol{\theta} \sim q_{e-1}$             ▷ *where $q_t = \mathcal{N}(\boldsymbol{\theta} \mid \mathbf{m}_t, \mathrm{diag}(\boldsymbol{\sigma}_t^2))$*
7:      $\widehat{\mathbf{h}} \leftarrow \widehat{\mathrm{g}} \cdot (\boldsymbol{\theta} - \mathbf{m})/\boldsymbol{\sigma^2}$
8:      $\mathbf{g} \leftarrow \beta_1\mathbf{g} + (1 - \beta_1)\widehat{\mathbf{g}}$
9:      $\mathbf{h} \leftarrow \beta_2\mathbf{h} + (1 - \beta_2)\widehat{\mathbf{h}} + \dfrac{1}{2}(1 - \beta_2)^2(\mathbf{h} - \widehat{\mathbf{h}})^2/(\mathbf{h} + \delta)$
10:      $\overline{\mathbf{g}} \leftarrow \mathbf{g}/(1 - \beta_1^e)$
11:      $\mathbf{m} \leftarrow \mathbf{m} - \alpha_e(\overline{\mathbf{g}} + \delta\mathbf{m})/(\mathbf{h} + \delta)$
12:      $\boldsymbol{\sigma} \leftarrow 1/\sqrt{\lambda(\mathbf{h} + \delta)}$
13: **end for**

---

**Algorithm 2** FedIvon Algorithm

---

**Input:** Total communication rounds $R$, total clients $K$, clients' private datasets $\{\mathcal{D}_k\}_{k=1}^K$, initial model weight $\tilde{\boldsymbol{m}}_\mathbf{0}$, initial model Hessian $\tilde{\boldsymbol{h}}_\mathbf{0}$

1: **for** $r = 1$ to $R$ **do**
2:      Broadcast $\tilde{\boldsymbol{m}}_\boldsymbol{r}, \tilde{\boldsymbol{h}}_\boldsymbol{r}$ to all $K$ clients
3:      Randomly sample $k$ clients             ▷ *Update selected client models locally*
4:      **for** $i = 1$ to $k$ **do**
5:          $(\boldsymbol{m}_i, \boldsymbol{h}_i) \leftarrow \mathrm{CLIENT\_UPDATE}(D_i, \tilde{\boldsymbol{m}}_\boldsymbol{r}, \tilde{\boldsymbol{h}}_\boldsymbol{r})$
6:      **end for**

7:      Initialize $\tilde{\boldsymbol{m}}_{\boldsymbol{r+1}} \leftarrow 0, \tilde{\boldsymbol{h}}_{\boldsymbol{r+1}} \leftarrow 0$             ▷ *Aggregation of client models at server*
8:      **for** $i = 1$ to $k$ **do**
9:          $\tilde{\boldsymbol{h}}_{\boldsymbol{r+1}} \leftarrow \tilde{\boldsymbol{h}}_{\boldsymbol{r+1}} + \boldsymbol{h}_i * w[i]$             ▷ *$w[i]$ is relative dataset size $|\mathcal{D}_i|/|\mathcal{D}|$*
10:          $\tilde{\boldsymbol{m}}_{\boldsymbol{r+1}} \leftarrow \tilde{\boldsymbol{m}}_{\boldsymbol{r+1}} + (\boldsymbol{m}_i \odot \boldsymbol{h}_i) * w[i]$
11:      **end for**
12:      $\tilde{\boldsymbol{m}}_{\boldsymbol{r+1}} \leftarrow \tilde{\boldsymbol{m}}_{\boldsymbol{r+1}}/\tilde{\boldsymbol{h}}_{\boldsymbol{r+1}}$             ▷ *elementwise division; global weight and Hessian*
13: **end for**

**Output:** Global model weights and Hessian $(\tilde{\boldsymbol{m}}_\boldsymbol{R}, \tilde{\boldsymbol{h}}_\boldsymbol{R})$

---

Other aggregation strategies are also possible (Fischer et al., 2024) and we leave this for future work. Note that our aggregation strategy can also be seen as Fisher-weighted model merging (Daheim et al., 2023) where each client model is represented as the mean weights $\boldsymbol{m}_k$ and a Fisher matrix which depends on local posterior's variances $\boldsymbol{\sigma}_k^2$ (although model merging only computes the mean, not the covariance, and thus does not yield a global posterior distribution at the server).

The appendix C provides further details of IVON and its integration in our Bayesian FL setup.

Notably, FedIvon is appealing from two perspectives. It can be viewed an an efficient Bayesian FL algorithm offering the various benefits of the Bayesian approach, as well as a federated learning algorithm that easily incorporates second-order information during the training of the client models, while not incurring the usual overheads of second-order methods used by some FL algorithms (Bischoff et al., 2021a).

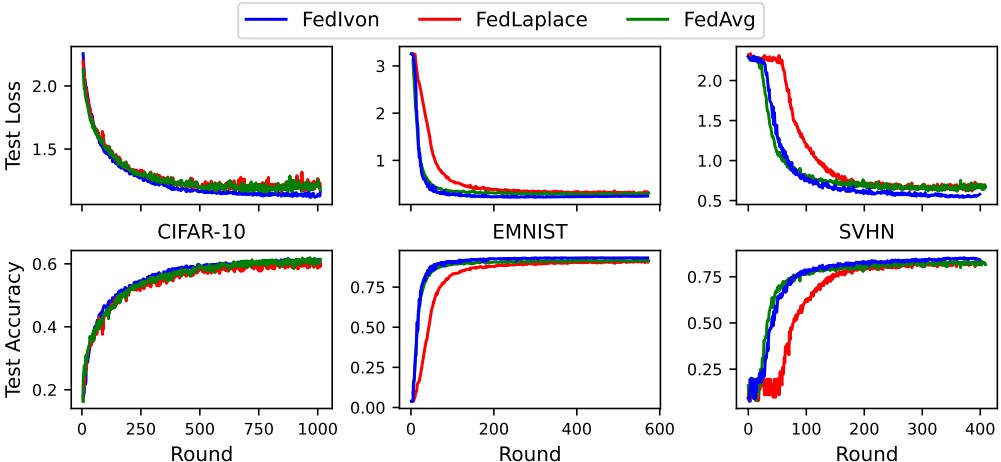

Figure 2: Loss and Accuracy of various methods vs rounds for the EMNIST, SVHN, and CIFAR-10 datasets.

## 6   Personalized Federated Learning

Personalized FL in FedIvon can be achieved straightforwardly. Similar to equation 3, the personalized loss function for each client $k$ is defined as,

$$\mathcal{L}_k(q) = \mathbb{E}_{q_k(\boldsymbol{\theta})}[\bar{\ell}_k(\boldsymbol{\theta})] + \beta \, \mathbb{D}_{KL}(q_k(\boldsymbol{\theta})\|p_k(\boldsymbol{\theta})). \tag{7}$$

Where $\beta \geq 0$ controls the level of personalization. The term $p_k(\boldsymbol{\theta})$ represents the prior distribution for client $k$. During each communication round, the posterior distribution from the server can be used as the prior $p_k(\boldsymbol{\theta})$ for the client. This setup enables clients to adapt the global model according to their local data characteristics while leveraging information from the global model.

When $\beta = 0$, the model becomes fully personalized, relying solely on the client's data without influence from the prior (i.e., no information from the server). Conversely, a higher value of $\beta$ incorporates more knowledge from the global server model into the client's learning process, balancing between personalization and shared global information. This framework provides a flexible mechanism to adapt client models according to their individual data while still benefiting from collective learning through the shared server posterior. We fixed $\beta = 1$ in all our pFL experiments.

## 7   Experiments: Standard FL

We experiment on three publicly available datasets: EMNIST (Cohen et al., 2017), SVHN (Netzer et al., 2011) and CIFAR-10 (Krizhevsky, 2009). EMNIST consists of 28x28 grayscale images of alphabets and digits (0-9) with a train and test split comprising 124800 and 20800 images respectively; however, in our experiments, we restrict to alphabets only. SVHN consists of 32x32 RGB images of house number plates categorized into 10 distinct classes, each corresponding to one of the ten digits. It has a train and test split of size 73252 and 26032 respectively. CIFAR-10 comprises 32x32 RGB images of objects classified into 10 classes with 50000 training images and 10000 test images.

In our experiments, We use Adam optimizer with `learning_rate=1e-3, weight_decay=2e-4` for FedAvg and FedLaplace method. IVON (Shen et al., 2024) optimizer is used for FedIvon with different hyperparameters given in Table 6. Linearly decaying learning rate is used in all the experiments.

We evaluate FedIvon in a challenging and realistic scenario involving heterogeneous data distribution among a large number of clients with each client having very few training examples. For each experiment, we consider a total of 200 clients with each client having a small private training set of less than 100 examples. To simulate non-i.i.d. data distribution, we randomly sample inputs from the training split, partition the sampled inputs into shards, and distribute shards among clients to create class-imbalanced training data

similar to (Chen & Chao, 2020). For a fair comparison, we use the same non-i.i.d. data split across clients for all the baseline methods and FedIvon. We follow the experimental setup of (Bhatt et al., 2023) and train customized CNN models on EMNIST, SVHN, and CIFAR-10 datasets. We compare our proposed method FedIvon with FedAvg (McMahan et al., 2017) (simple aggregation of client models at server) and FedLaplace (Liu et al., 2024) (using the Laplace's approximation to fit a Gaussian distribution to each client's local model followed by aggregation at the server). FedAvg serves as a baseline to emphasize the importance of uncertainty quantification without compromising on the performance while FedLaplace serves as a competitive baseline to evaluate FedIvon's predictive uncertainty measures. For all the baselines and FedIvon, we run the federated algorithm for 2000 communication rounds, selecting a randomly sampled 5% i.e., 10 clients per round. We train each client's model locally for 2 epochs using a batch size of 32. We provide further details on hyperparameters, model architectures, and split in the appendix.

## 7.1 Classification Task

We train a classification model in FL setting using all the methods and report the results in Table 1. We evaluate all trained models' performance (accuracy and negative log-likelihood) on the test split and use metrics such as Expected Calibration Error (ECE) and Brier score to quantify predictive uncertainty. In our results, FedIvon@mean denotes point estimate based predictions evaluated at the mean of IVON posterior and FedIvon corresponds to Monte Carlo averaging with 500 samples from the posterior.

| Metric | Dataset | FedAvg | FedLaplace | FedIvon@mean | FedIvon |
|---|---|---|---|---|---|
| ACC($\uparrow$) | EMNIST | 91.66 | 91.33 | **93.14** | 93.09 |
| | CIFAR-10 | 62.25 | 61.80 | **62.92** | 62.54 |
| | SVHN | 82.14 | 81.99 | 84.54 | **84.76** |
| ECE($\downarrow$) | EMNIST | 0.0405 | 0.0381 | 0.0349 | **0.0188** |
| | CIFAR-10 | 0.0981 | 0.1072 | 0.0983 | **0.0312** |
| | SVHN | 0.0311 | 0.0211 | 0.0241 | **0.0148** |
| NLL($\downarrow$) | EMNIST | 0.3355 | 0.3255 | 0.2821 | **0.2341** |
| | CIFAR-10 | 1.199 | 1.233 | 1.1500 | **1.0790** |
| | SVHN | 0.6857 | 0.6423 | 0.5624 | **0.5303** |
| BS($\downarrow$) | EMNIST | 0.1303 | 0.1314 | 0.1075 | **0.1019** |
| | CIFAR-10 | 0.5191 | 0.5284 | 0.5114 | **0.5021** |
| | SVHN | 0.2640 | 0.2627 | 0.2256 | **0.2210** |

Table 1: Test accuracy (ACC), Expected Calib. Error (ECE), Negative Log Likelihood (NLL), and Brier Score (BS)

As shown in Table 1, FedIvon outperforms all the baselines and yields the best test performance and calibration scores. FedIvon leverages the Improved Variational Online Newton (IVON) method to approximate the Hessian by continuous updates throughout the training. We also show the convergence of all the methods on all the datasets in Figure 2. As observed, FedIvon exhibits slightly slower improvements in the early training phase as compared to other baselines but soon outperforms them owing to its improved Hessian approximation as training progresses. Moreover, unlike FedLaplace which fits Gaussian distribution to the client's model using Laplace approximation evaluated at MAP estimate, FedIvon approximates the Hessian over the entire course of its training, resulting in much better predictive uncertainty estimates. Figure 3 shows the reliability diagrams for FedAvg, FedLaplace, and FedIvon on the CIFAR-10 and EMNIST datasets. This clearly shows that FedIvon yields better-calibrated predictions than both FedAvg and FedLaplace. For other dataset reliability diagrams are shown in the appendix E.

## 7.2 Out-of-Distribution Detection Task

Predictive uncertainty of the model plays a crucial role in uncertainty-driven tasks such as OOD detection and active learning. We evaluate FedIvon and the baselines for distinguishing OOD inputs from in-distribution inputs using their predictive uncertainty. Given any input **x**, the predictive uncertainty of the model's output is given by its Shannon entropy and is used to filter OOD inputs.

| Models | EMNIST | CIFAR-10 | SVHM |
|---|---|---|---|
| FedAvg | 0.8910 | **0.7896** | 0.7975 |
| FedLaplace | 0.8297 | 0.7513 | 0.8222 |
| FedIvon | **0.9032** | 0.7662 | **0.8233** |

Table 2: AUROC ($\uparrow$) score for OOD/in-domain data detection

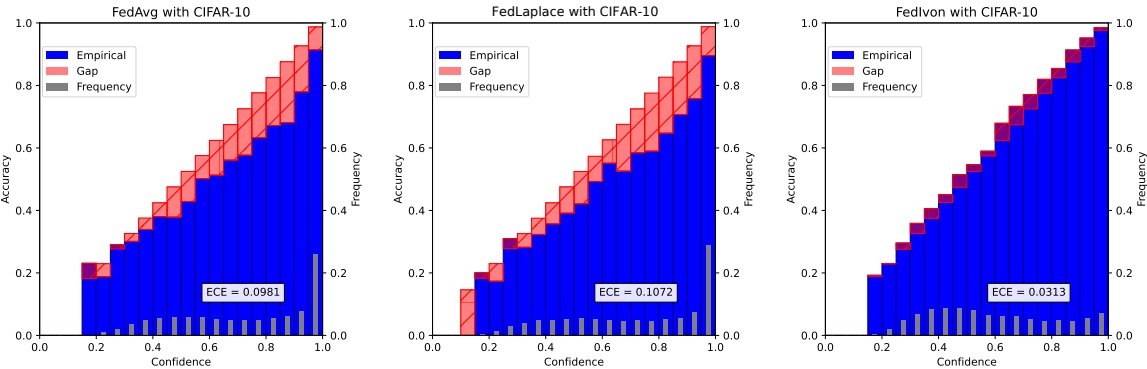

Figure 3: Reliability diagrams for CIFAR-10 experiments (left: FedAvg, center: Fedlaplace, right: FedIvon).

We simulate this task by randomly sampling 5000 images from the OOD dataset and mixing it with an equal number of randomly sampled inputs from the test split of the training dataset.

Specifically, we use EMNIST, CIFAR-10, and SVHN as the OOD dataset for the models trained on EMNIST, SVHN, and CIFAR-10 respectively. We report the AUROC (area under the ROC curve) metric for all the methods on all the datasets in Table 2 which shows that FedIvon achieves better or competitive AUROC scores as compared to the other baselines.

### 7.3 Effect of the number of local epochs.

We empirically investigate the impact of varying the number of local epochs $E$ on the convergence behaviour of the different methods at the server. Figure 4 shows the convergence trajectories for several values of $E$. When $E = 1$, FedIvon converges more slowly than FedAvg, and FedLaplace converges even more slowly. For FedIvon, this slower initial progress can be attributed to the fact that gradients are computed using stochastic weight samples; at initialisation, these samples yield noisier gradient estimates, which weakens the quality of early updates. FedLaplace, which requires the computation of a MAP estimate, also suffers at $E = 1$: a single epoch of training produces a suboptimal MAP point and therefore leads to slower improvement. When the number of local epochs is increased to $E = 2$, all methods show faster convergence compared with the $E = 1$ setting. Additional local iterations provide better gradient estimates for FedIvon and a more accurate MAP estimate for FedLaplace, leading to improved server-side progress. After a small number of communication rounds, FedIvon surpasses both FedAvg and FedLaplace, reflecting its ability to refine its variational updates as the global posterior becomes more informative.

However, increasing the number of local epochs beyond $E = 2$ leads to unstable training behaviour for FedIvon (and, more generally, for federated posterior–learning algorithms in which the server posterior is reused as a prior for the clients in subsequent rounds). Large values of $E$ make each client's posterior highly concentrated (see Algorithm 1; seeing the same data multiple times naturally increases posterior precision). When these high-precision client posteriors are aggregated at the server and the resulting global posterior is sent back to the clients as the prior for the next round, this sharp prior can dominate the local likelihood. In this regime, the client update behaves as if it were heavily regularized around the global mean, which limits the client's ability to learn from its own data. As this prior–likelihood imbalance is repeatedly amplified across rounds, the global posterior becomes progressively sharper and the overall training dynamics can become unstable, particularly under highly non-i.i.d. conditions. For this reason, we restrict the number of local epochs to $E = 2$ for all the experiments, which provides a balance between improved local optimization and stable global aggregation.

## 8 Communication and computational cost

Let $P$ denote the number of model parameters. In each local step, FedIvon draws a single Monte Carlo sample $\theta = m + \sigma \odot \varepsilon$ and performs one forward and one backward pass to obtain a stochastic gradient.

The subsequent IVON updates (momentum, diagonal curvature $h$, and mean/variance updates) require only elementwise $\mathcal{O}(P)$ operations. Thus, the per–minibatch computational cost is $C_{\text{FedIvon}}(B) = C_{\text{fw}} + C_{\text{bw}} + \mathcal{O}(P)$, which is the same asymptotic cost as FedAvg/Adam. The difference between FedIvon and FedAvg is only by a constant factor: FedIvon performs the same forward–backward pass but includes several additional $\mathcal{O}(P)$ elementwise updates. In practice, this overhead is small because the forward–backward pass dominates the runtime. For client $k$, the cost per communication round is $C^{(k)}_{\text{FedIvon, round}} = \frac{E|D_k|}{B}\big(C_{\text{fw}} + C_{\text{bw}} + \mathcal{O}(P)\big)$, identical in scaling to FedAvg and differing only by the small constant factor above.

Communication per round is linear in model size. Each client uploads its posterior mean and diagonal precision (two vectors in $\mathbb{R}^P$) and downloads the corresponding global quantities, for a total of $4P$ scalars per round. This is a factor of two more than FedAvg, which transmits only a single parameter vector in each direction. In typical federated-learning settings, where communication budgets are dominated by transmitting full model weights, doubling the number of transmitted vectors (from $P$ to $2P$) does not affect feasibility, the order of communication remains $\Theta(P)$.

Table 3 reports the empirical communication and computation costs per communication round for all methods. `Communication/round` denotes the total amount of data transmitted between server and clients (uplink + downlink) in megabytes, averaged over rounds and participating clients. `FLOPs/round` is the average number of floating-point operations per round (estimated via PyTorch's profiler), and `Runtime/round` is the corresponding average wall-clock time in seconds.

As expected, FedAvg is the most efficient baseline in all three metrics, since it only communicates model weights and performs a single forward-backward pass per batch. Among the Bayesian methods, FedLaplace and FedIvon incur roughly twice the communication cost of non-Bayesian method (FedAvg), because they must transmit both the mean parameters and a diagonal covariance between server and clients. In terms of FLOPs and wall-clock runtime, FedLaplace is the most expensive due to the additional passes required to form a Laplace approximation, whereas FedIvon remains close to FedAvg. Overall, these results indicate that FedIvon achieves Bayesian uncertainty quantification with a runtime overhead that is comparable to standard first-order federated optimizers, at the cost of a twice increase in communication.

| Method | CC (MB) | FLOPs ($\times 10^9$) | Runtime (s) |
|---|---|---|---|
| FedAvg | 9.7 | 8.34 | 12.94 |
| FedLaplace | 19.4 | 16.71 | 21.07 |
| FedIvon | 19.4 | 8.48 | 15.05 |

Table 3: Per-round communication cost (CC), computational cost (FLOPs), and wall-clock runtime for different federated learning algorithms. Values are averaged over global rounds and participating clients. The model and dataset were fixed across methods, so the numbers are best interpreted relatively.

## 9 Experiments: Personalized FL

For personalized FL experiments, we focus on two types of data heterogeneity on the clients similar to Zhu et al. (2023) for classification task. We compare our approach `FedIvon` against personalized federated baselines (`pFedME` (T Dinh et al., 2020), `pFedBayes` (Zhang et al., 2022), and `pFedVEM` (Zhu et al., 2023)).

- **Class distribution skew**: In class distribution skew, clients have data from only a limited set of classes. To simulate this, we use the CIFAR-10 dataset and assign each client data from a random selection of 5 out of the 10 classes.

- **Class concept drift**: To simulate class concept drift, we use the CIFAR-100 dataset, which includes 20 superclasses, each containing 5 subclasses. For each client, we randomly select one subclass from each superclass (1 out of 5). The client's local data is then drawn exclusively from these selected subclasses, creating a shift in label concepts across clients. We define the classification task as predicting the 20 superclass.

To model data-quantity disparity, we randomly divide the training set into partitions of varying sizes by uniformly sampling slice indices, then assign each partition to a different client.

| Clients | Model | Method | | | | | |
|---------|-------|-------|--------|-----------|---------|---------------|----------|
| | | Local | pFedME | pFedBayes | pFedVEM | **FedIvon@mean** | **FedIvon** |
| 50 | PM | $56.9 \pm 0.1$ | $72.3 \pm 0.1$ | $71.4 \pm 0.3$ | $73.2 \pm 0.2$ | $74.4 \pm 0.3$ | $\textbf{75.5} \pm \textbf{0.4}$ |
| | GM | - | $56.6 \pm 1.0$ | $52.0 \pm 1.0$ | $56.0 \pm 0.4$ | $\underline{67.1 \pm 1.0}$ | $\textbf{67.8} \pm \textbf{1.6}$ |
| 100 | PM | $52.1 \pm 0.1$ | $71.4 \pm 0.2$ | $68.5 \pm 0.3$ | $\underline{71.9 \pm 0.1}$ | $71.7 \pm 0.3$ | $\textbf{72.6} \pm \textbf{0.2}$ |
| | GM | - | $60.1 \pm 0.3$ | $53.2 \pm 0.7$ | $60.1 \pm 0.2$ | $\underline{68.4 \pm 0.2}$ | $\textbf{69.2} \pm \textbf{0.2}$ |
| 200 | PM | $46.6 \pm 0.1$ | $68.5 \pm 0.2$ | $64.6 \pm 0.2$ | $\underline{70.1 \pm 0.3}$ | $69.7 \pm 0.7$ | $\textbf{70.8} \pm \textbf{0.4}$ |
| | GM | - | $58.7 \pm 0.2$ | $51.4 \pm 0.3$ | $59.4 \pm 0.3$ | $\underline{68.2 \pm 0.3}$ | $\textbf{68.7} \pm \textbf{0.3}$ |

Table 4: Comparison of Personalized FL Methods on CIFAR-10

| Clients | Model | Method | | | | | |
|---------|-------|-------|--------|-----------|---------|---------------|----------|
| | | Local | pFedME | pFedBayes | pFedVEM | **FedIvon@mean** | **FedIvon** |
| 50 | PM | $34.3 \pm 0.2$ | $52.5 \pm 0.5$ | $49.6 \pm 0.3$ | $61.0 \pm 0.4$ | $65.4 \pm 0.7$ | $\textbf{66.7} \pm \textbf{0.8}$ |
| | GM | - | $47.9 \pm 0.5$ | $42.5 \pm 0.5$ | $52.8 \pm 0.4$ | $\underline{63.2 \pm 0.5}$ | $\textbf{63.8} \pm \textbf{0.7}$ |
| 100 | PM | $27.6 \pm 0.3$ | $47.6 \pm 0.5$ | $46.5 \pm 0.2$ | $56.2 \pm 0.4$ | $63.2 \pm 0.5$ | $\textbf{63.5} \pm \textbf{0.6}$ |
| | GM | - | $45.1 \pm 0.3$ | $41.3 \pm 0.3$ | $52.3 \pm 0.4$ | $\underline{62.1 \pm 0.5}$ | $\textbf{62.4} \pm \textbf{0.6}$ |
| 200 | PM | $22.2 \pm 0.2$ | $41.6 \pm 1.8$ | $40.1 \pm 0.3$ | $51.1 \pm 0.6$ | $56.1 \pm 0.6$ | $\textbf{56.5} \pm \textbf{0.5}$ |
| | GM | - | $41.5 \pm 1.6$ | $37.4 \pm 0.3$ | $49.2 \pm 0.5$ | $\underline{55.5 \pm 0.6}$ | $\textbf{55.7} \pm \textbf{0.7}$ |

Table 5: Comparison of Personalized FL Methods on CIFAR100

## 9.1 Setup

We evaluate our approach in 3 different settings: number of clients $K \in \{50, 100, 200\}$. We followed the same model architectures as the prior work (Zhu et al., 2023). A simple 2-convolution layered-based model is used for CIFAR-10, while a deeper model having 6 convolution layers is used for the CIFAR-100 dataset. We assess both a personalized model (PM) and a global model (GM) at the server. The PMs are evaluated using test data that matches the labels (for label distribution skew) or subclasses (for label concept drift) specific to each client, while the GM is evaluated on the entire test set. All experiments are repeated 3 times, using the same set of 3 random seeds for data generation, parameter initialization, and client sampling. For each client, the Ivon hyperparameters used are given in Table 7.

## 9.2 Results

Table 4 and 5 presents results on CIFAR-10 and CIFAR-100 datasets respectively, which are used to simulate different types of data heterogeneity in federated learning. Here local method refers to client only training without communication to server. CIFAR-10 represents class distribution skew, where each client has data from a limited set of classes, while CIFAR-100 represents class concept drift, where each client has data from distinct subclasses within superclasses. For both datasets, we evaluate client's average accuracy (personalized model) and server accuracy (global model) across varying client counts (50, 100, and 200). FedIvon uses 64 Monte Carlo samples to perform Monte Carlo averaging. On the other hand, FedIvon@mean uses a point estimate using mode of the posterior.

On CIFAR-10, FedIvon achieves similar client accuracy to pFedVEM, indicating both methods perform well under class distribution skew for individual clients. However, in server accuracy, FedIvon shows a notable improvement over pFedVEM and other methods, highlighting FedIvon's strength in aggregating data from heterogeneous clients into an accurate global model.

On CIFAR-100, which represents class concept drift, FedIvon demonstrates significant improvements over all other methods in both client's average accuracy and server accuracy. This performance advantage in both personalized and global evaluations suggests that FedIvon is well-suited to handling concept drift, achieving higher accuracy for individual clients and in the global model. Overall, FedIvon consistently outperforms other methods in both accuracy metrics on CIFAR-10 and CIFAR-100, underscoring its robustness across different data heterogeneity scenarios.

## 10   Limitations

One limitation of FedIvon is that it approximates the global posterior with a diagonal Gaussian. If the true global posterior is not well approximated by a single diagonal Gaussian, for example, if it exhibits strong parameter correlations, pronounced non-Gaussianity, or multimodal structure then this variational family cannot faithfully capture the geometry of the posterior. In such cases, the resulting approximation may distort epistemic uncertainty by misallocating variance across parameters and ignoring important covariance structure, which can in turn lead to miscalibrated predictive uncertainties, especially in regimes with strong client disagreement or significant distribution shift.

FedIvon also introduces additional resource overhead compared to FedAvg. On the client side, IVON requires storing extra state (Hessian), which increases memory by a small constant factor in our experiments but would be more substantial for very large models. Communication per round is roughly doubled, since each client sends and receives both mean and diagonal Hessian vectors instead of a single parameter vector. Although this overhead remains practical for the moderate-size CNNs and vision benchmarks considered here, it would be more significant in large-scale settings such as LLM training.

## 11   Conclusion

We presented a new Bayesian Federated Learning (FL) method that reduces the computational and communication overhead typically associated with Bayesian approaches. Our method uses an efficient second-order optimization technique for variational inference, achieving computational efficiency similar to first-order methods like Adam while still providing the benefits of Bayesian FL, such as uncertainty estimation and model personalization. We showed that our approach improves predictive accuracy and uncertainty estimates compared to both Bayesian and non-Bayesian FL methods. Additionally, our method naturally supports personalized FL by allowing clients to use the server's posterior as a prior for learning their own models.

**Acknowledgment** P.R. acknowledges support from Google Research and Satya & Rao Remala Foundation. SP and SS acknowledge support from Visvesvaraya fellowship.

## Broader Impact

FedIvon is a Bayesian personalized federated learning method, and its deployment raises several considerations that we briefly discuss here.

**Fairness across clients.** By design, personalized federated learning can yield heterogeneous model quality across clients. Clients with more or higher-quality data may obtain substantially better personalized posteriors than those with limited or noisy data. In settings where model performance has direct societal or economic consequences, this may amplify existing inequalities between users or institutions. FedIvon does not explicitly address such fairness concerns; incorporating fairness-aware objectives or constraints into Bayesian personalized federated learning is an important direction for future work.

**Reliability of uncertainty estimates.** A key motivation for FedIvon is improved uncertainty calibration, and we observe benefits on the benchmarks considered. However, miscalibrated uncertainty can be harmful in high-stakes applications if used to drive decisions. In particular, the diagonal posterior approximation and limited per-client data may lead to overconfidence on out-of-distribution inputs or on minority groups underrepresented in the participating clients. Our experiments are restricted to vision benchmarks in controlled settings, and we do not claim that FedIvon provides robust uncertainty guarantees in safety-critical domains. Stress-testing under distribution shift and demographic imbalance remains necessary before deployment in such contexts.

**Privacy and security.** Although federated learning keeps raw data on-device, transmitting model updates or posterior parameters can still leak information about local data. FedIvon shares client-specific posterior means and variances, which may, in principle, be exploited by a malicious server or adversary to infer properties of client data. We do not incorporate differential privacy, secure aggregation, or other formal privacy mechanisms, and therefore do not claim any formal privacy guarantees. Combining FedIvon with established privacy-preserving techniques is an important prerequisite for use in sensitive applications.

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

# A  Theoretical insight and stability considerations

We do not provide a full convergence proof for FedIvon in the federated, non-convex setting. Instead, we sketch simple conditions under which the IVON-based client updates and the FedIvon aggregation lead to a qualitatively stable fixed-point iteration.

Single-client IVON Algorithm 1 maintains a diagonal-Gaussian posterior

$$q_t(\theta) = \mathcal{N}(m_t, \Lambda_t^{-1}), \qquad \Lambda_t = \lambda\,(h_t + \delta),$$

where $\lambda = |D|$, $h_t \geq 0$ is the diagonal Hessian estimate and $\delta > 0$ is the damping used in the update of $\sigma$. At each step,

$$\hat{g}_t \leftarrow \nabla\bar{\ell}(\theta_t), \qquad \hat{h}_t \leftarrow \hat{g}_t \odot \frac{\theta_t - m_t}{\sigma_t^2},$$

and $h_t$ is updated as:

$$h_{t+1} \;\leftarrow\; (1-\rho)h_t + \rho\left[\hat{h}_t + \frac{\rho}{2}\frac{(h_t - \hat{h}_t)^2}{h_t + s_0/\lambda}\right],$$

for some $\rho \in (0,1)$ and $s_0 > 0$. Shen et al. (2024) show that if $h_0 > 0$, then this update guarantees $h_t > 0$ for all $t$. Assuming the curvature statistics are bounded, $|\hat{h}_t| \leq H_{\max}$, the EMA implies

$$0 < h_{\min} \leq h_t \leq h_{\max} < \infty.$$

With $\delta > 0$, the diagonal precision

$$\Lambda_t = \lambda(h_t + \delta)$$

satisfies

$$\lambda\delta \;\leq\; \Lambda_t \;\leq\; \lambda(h_{\max} + \delta) \quad \forall t,$$

so $\Lambda_t$ is uniformly positive and bounded. The mean update is

$$m_{t+1} \;\leftarrow\; m_t - \alpha_t\,\frac{\tilde{g}_t + \delta m_t}{h_t + \delta},$$

where $\tilde{g}_t$ is the bias-corrected gradient momentum. This can be written as a preconditioned gradient step

$$m_{t+1} = m_t - \alpha_t\,\Lambda_t^{-1}\,g_t',$$

for some effective gradient $g_t'$. Because $\Lambda_t^{-1}$ is also bounded,

$$0 < \frac{1}{\lambda(h_{\max} + \delta)} \leq \Lambda_t^{-1} \leq \frac{1}{\lambda\delta} < \infty\,.$$

Under standard smoothness assumptions and small $\alpha_t$, this yields a stable preconditioned SGD iteration for the single-client IVON update.

**FedIvon aggregation with multiple clients.** In FedIvon, client $k$ returns

$$q_k(\theta) = \mathcal{N}(m_k, \Lambda_k^{-1}), \qquad \Lambda_k = \lambda_k(h_k + \delta),$$

with the same IVON bounds $0 < h_{\min} \leq h_k \leq h_{\max}$. Therefore

$$\lambda_k \delta \leq \Lambda_k \leq \lambda_k(h_{\max} + \delta) \quad \forall k.$$

The server aggregates posteriors in natural-parameter space:

$$\Lambda_{\text{server}}^{(t)} = \sum_{k=1}^{K} w_k \Lambda_k^{(t)}, \qquad m_{\text{server}}^{(t)} = (\Lambda_{\text{server}}^{(t)})^{-1} \sum_{k=1}^{K} w_k \Lambda_k^{(t)} m_k^{(t)},$$

where $w_k = |D_k| / \sum_j |D_j|$ and $K \geq 2$. By convexity we obtain uniform bounds

$$\Lambda_{\min} := \min_k \{\lambda_k \delta\} \ \leq \ \Lambda_{\text{server}}^{(t)} \ \leq \ \max_k \{\lambda_k(h_{\max} + \delta)\} =: \Lambda_{\max},$$

for all communication rounds $t$. Hence the global precision is also strictly positive and bounded, and the global covariance never degenerates or explodes. In the next round, each client uses $q_{\text{server}}$ as its prior and applies a finite number of IVON steps. With a small number of local epochs $E$, each client performs only a single approximate Bayesian update per round, so that all $\Lambda_k$ remain within the same bounded interval. The combined client–server mapping on $(m, \Lambda)$ is therefore a stable fixed-point iteration over a compact subset of the diagonal-Gaussian family. A full convergence analysis for non-iid, multi-client, non-convex models is beyond the scope of this work, but these boundedness properties explain why FedIvon remains numerically stable in our experiments.

# B  Proof of claim 1

Let $K$ clients hold disjoint datasets $\mathcal{D}_1, \ldots, \mathcal{D}_K$, and assume a common prior $p(\boldsymbol{\theta})$. Define the global dataset $\mathcal{D} = \biguplus_{k=1}^{K} \mathcal{D}_k$. By Bayes' rule and conditional independence, the global posterior is

$$p(\boldsymbol{\theta}|\mathcal{D}) \propto p(\boldsymbol{\theta}) \, p(\mathcal{D}_1, \ldots, \mathcal{D}_K | \boldsymbol{\theta}) \propto p(\boldsymbol{\theta}) \prod_{k=1}^{K} p(\mathcal{D}_k | \boldsymbol{\theta}). \tag{8}$$

Each local posterior can be expressed as

$$p(\boldsymbol{\theta}|\mathcal{D}_k) \propto p(\boldsymbol{\theta}) \, p(\mathcal{D}_k | \boldsymbol{\theta}). \tag{9}$$

Taking the product over all clients yields

$$\prod_{k=1}^{K} p(\boldsymbol{\theta}|\mathcal{D}_k) \propto p(\boldsymbol{\theta})^K \prod_{k=1}^{K} p(\mathcal{D}_k | \boldsymbol{\theta}), \tag{10}$$

which implies the exact relationship

$$p(\boldsymbol{\theta}|\mathcal{D}) \propto \frac{\prod_{k=1}^{K} p(\boldsymbol{\theta}|\mathcal{D}_k)}{p(\boldsymbol{\theta})^{K-1}}. \tag{11}$$

Equation (11) gives the *exact* global posterior (up to normalization). However, it assigns equal importance to each local posterior, regardless of dataset size. In practice, client datasets may be imbalanced ($|\mathcal{D}_k|$ varies), so we instead adopted a *weighted approximation*, following prior works (Liu et al., 2024), to better reflect the contribution of each client.

**Weighted Approximation.** We define normalized weights $w_k = \frac{|\mathcal{D}_k|}{|\mathcal{D}|}$ and approximate the global posterior as

$$p(\boldsymbol{\theta}|\mathcal{D}) \approx \prod_{k=1}^{K} q(\boldsymbol{\theta}|\mathcal{D}_k)^{w_k}, \tag{12}$$

where $q(\boldsymbol{\theta}|\mathcal{D}_k) \approx p(\boldsymbol{\theta}|\mathcal{D}_k)$ denotes the local approximate posterior. Equation (12) serves as an approximation to the global posterior in Eq. (8). The weighting by $w_k$ ensures that clients with larger datasets contribute proportionally more and implicitly prevents the overcounting of the prior.

Each local approximation can be written as

$$q(\boldsymbol{\theta}|\mathcal{D}_k) \approx \frac{1}{Z_k} p(\boldsymbol{\theta}) p(\mathcal{D}_k|\boldsymbol{\theta}),$$

which implies

$$q(\boldsymbol{\theta}|\mathcal{D}_k)^{w_k} \approx p(\boldsymbol{\theta})^{w_k} \, p(\mathcal{D}_k|\boldsymbol{\theta})^{w_k}.$$

Substituting this into Eq. (12), we obtain

$$\prod_{k=1}^{K} q(\boldsymbol{\theta}|\mathcal{D}_k)^{w_k} \approx p(\boldsymbol{\theta})^{\sum_{k=1}^{K} w_k} \prod_{k=1}^{K} p(\mathcal{D}_k|\boldsymbol{\theta})^{w_k}.$$

Since $\sum_{k=1}^{K} w_k = 1$, this simplifies to

$$\prod_{k=1}^{K} q(\boldsymbol{\theta}|\mathcal{D}_k)^{w_k} \propto p(\boldsymbol{\theta}) \prod_{k=1}^{K} p(\mathcal{D}_k|\boldsymbol{\theta})^{w_k}. \tag{13}$$

Equation (13) has the same form as the exact global posterior in Eq. (8), except that each likelihood term is raised to the power $w_k$. Thus, Eq. (12) provides an *approximation* to the true posterior, where each client's contribution is balanced by the fraction of its local data.

**Gaussian Case.** If each local posterior is Gaussian, $q(\boldsymbol{\theta}|\mathcal{D}_k) = \mathcal{N}(\boldsymbol{\mu}_k, \boldsymbol{\Sigma}_k)$, then the weighted product in Eq. (12) is also Gaussian (Bishop, 2006):

$$\prod_{k=1}^{K} \mathcal{N}(\boldsymbol{\mu}_k, \boldsymbol{\Sigma}_k)^{w_k} \propto \mathcal{N}(\boldsymbol{\mu}_S, \boldsymbol{\Sigma}_S),$$

with parameters

$$\boldsymbol{\Sigma}_S^{-1} = \sum_{k=1}^{K} w_k \boldsymbol{\Sigma}_k^{-1}, \tag{14}$$

$$\boldsymbol{\mu}_S = \boldsymbol{\Sigma}_S \left( \sum_{k=1}^{K} w_k \boldsymbol{\Sigma}_k^{-1} \boldsymbol{\mu}_k \right). \tag{15}$$

## C   More details on IVON

Computing exact gradients in equation 4 and 5 is difficult due to the expectation term in $\mathcal{L}_k(q)$. A naïve way to optimize is to use stochastic gradient estimators. However, these approaches are not very scalable due to the high variance in the gradient estimates. Using natural gradients, Khan & Lin (2017) gave improved gradient based update equations for the variational parameters and they call this approach Natural Gradient

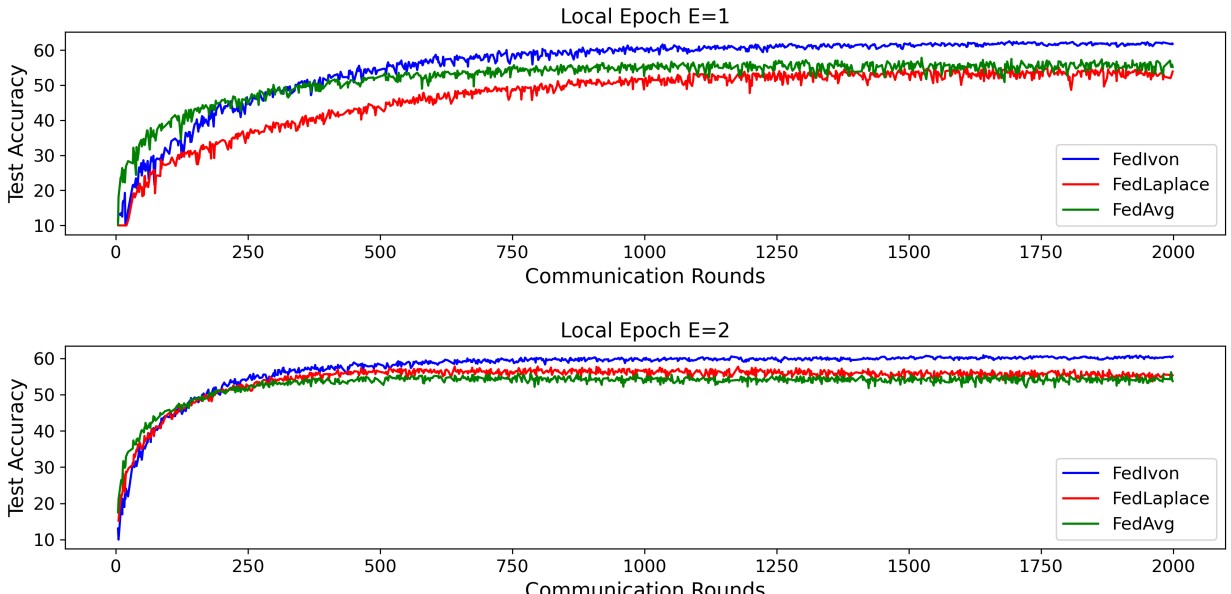

Figure 4: Convergence of all the methods on CIFAR-10 dataset with varying local epochs

VI (NGVI). The major difference between NVGI and original update equations is that learning rate is now adapted by the variance $\boldsymbol{\sigma}_k^{t+1}$ which makes these updates similar to Adam.

$$\textbf{NVGI: } \boldsymbol{m}_k^{t+1} = \boldsymbol{m}_k^t + \beta^t \boldsymbol{\sigma}_k^{2^{t+1}} \odot [\hat{\nabla}_{\boldsymbol{m}_k} \mathcal{L}_k(q)]$$
$$\boldsymbol{\sigma}_k^{-2^{t+1}} = \boldsymbol{\sigma}_k^{-2^t} - 2\beta^t [\hat{\nabla}_{\boldsymbol{\sigma}_k^2} \mathcal{L}_k(q)]$$

Further, Khan et al. (2018) showed that the NVGI update equations can be written in terms of scholastic gradient and Hessian of $\boldsymbol{\theta}$, where $\boldsymbol{\sigma}_k^{2^t} = [N(\boldsymbol{h}_k^t + \lambda)]^{-1}$. The vector $\boldsymbol{h}_k^t$ contains an online estimate of diagonal Hessian. This approach called Variational Online Newton (VON) is similar to NGVI except that it does not require the gradients of the variational objective.

$$\textbf{VON: } \boldsymbol{m}_k^{t+1} = \boldsymbol{m}_k^t - \beta^t \frac{\hat{\boldsymbol{g}}(\boldsymbol{\theta}^t) + \lambda \boldsymbol{m}_k^t}{\boldsymbol{h}_k^{t+1} + \lambda}$$
$$\boldsymbol{h}_k^{t+1} = (1 - \beta^t)\boldsymbol{h}_k^t + \beta^t \text{diag}[\hat{\nabla}_{\boldsymbol{\theta}\boldsymbol{\theta}}^2 \bar{\ell}_k(\boldsymbol{\theta}^t)]$$

In the update of VON for non-convex objective functions, the Hessian can be negative which might make $\boldsymbol{\sigma}_k^t$ negative, and break VON. To mitigate this issue Khan et al. (2018) used a Generalized Gauss-Newton (GGN) approximation of Hessian which is always positive. This method is called VOGN.

$$\nabla_{\theta_j \theta_j}^2 \bar{\ell}_k(\boldsymbol{\theta}^t) \approx \frac{1}{M} \sum_{i \in \mathcal{M}} \left[\nabla_{\theta_j} \ell_k^i(\boldsymbol{\theta}^t)\right]^2 := \hat{h}_j(\boldsymbol{\theta})$$

$$\textbf{VOGN: } \boldsymbol{m}_k^{t+1} = \boldsymbol{m}_k^t - \beta^t \frac{\hat{\boldsymbol{g}}(\boldsymbol{\theta}^t) + \lambda \boldsymbol{m}_k^t}{\boldsymbol{h}_k^{t+1} + \lambda}$$
$$\boldsymbol{h}_k^{t+1} = (1 - \beta^t)\boldsymbol{h}_k^t + \beta^t \hat{h}_j(\boldsymbol{\theta}^t)$$

VOGN Khan et al. (2018) improves these equations where Gauss Newton estimation is used instead of Hessian which gives similar update equations as the Adam optimizer. However, it still uses per-sample

squaring which is costly as compared to Adam.

$$\textbf{IVON:} \; \widehat{\mathbf{h}}_k^t = \widehat{\nabla}\bar{\ell}_k(\boldsymbol{\theta}) \cdot \frac{\boldsymbol{\theta} - \mathbf{m}_k^t}{\boldsymbol{\sigma}_k^{2^t}}$$

$$\mathbf{h}_k^{t+1} = (1 - \rho)\mathbf{h}_k^t + \rho\widehat{\mathbf{h}}_k^t + \frac{1}{2}\rho^2 \frac{(\mathbf{h}_k^t - \widehat{\mathbf{h}}_k^t)^2}{(\mathbf{h}_k^t + s_0/\lambda)}$$

Further, Shen et al. (2024) improved these update equations and provided much more efficient update equations similar to Adam optimizer, which is essentially the improved variational online Newton (IVON) algorithm Shen et al. (2024).

## D    Ivon Hyperparameter

| params | SVHN | EMNIST | CIFAR-10 |
|---|---|---|---|
| initial learning rate | 0.1 | 0.1 | 0.1 |
| final learning rate | 0.01 | 0.01 | 0.01 |
| weight decay | 2e-4 | 2e-4 | 2e-4 |
| batch size | 32 | 32 | 32 |
| ESS ($\lambda$) | 5000 | 5000 | 5000 |
| initial hessian ($h_0$) | 2.0 | 5.0 | 1.0 |
| MC sample while training | 1 | 1 | 1 |
| MC samples while test | 500 | 500 | 500 |

Table 6: Ivon Hyperparameters for FL experiments

## E    Reliability diagrams for FL experiments

Figures 3 and 5 show the reliability diagrams for CIFAR-10 and EMNIST experiments, respectively. The diagrams indicate that Fedivon has better-calibrated predictions compared to FedAvg and FedLaplace, as shown by its lower Expected Calibration Error (ECE).

## F    Client data distribution in FL experiments

Figure 6 illustrates the data distribution among clients used in the FL experiments. Each client has a highly imbalanced dataset, with the number of samples per client ranging from 5 to 32. Additionally, each client's dataset is limited to only a subset of classes, further emphasizing the non-IID nature of the data. This experimental setup poses significant challenges for training a robust global server model, as the limited and

| params | CIFAR-10 | CIFAR-100 |
|---|---|---|
| initial learning rate | 0.1 | 0.1 |
| final learning rate | 0.001 | 0.001 |
| weight decay | 1e-3 | 1e-3 |
| batch size | 32 | 32 |
| expected sample size ($\lambda$) | 10000 | 10000 |
| initial hessian ($h_0$) | 1.0 | 1.0 |
| MC sample while training | 1 | 1 |
| MC samples while test | 64 | 64 |

Table 7: Ivon Hyperparameters for personalized FL experiments

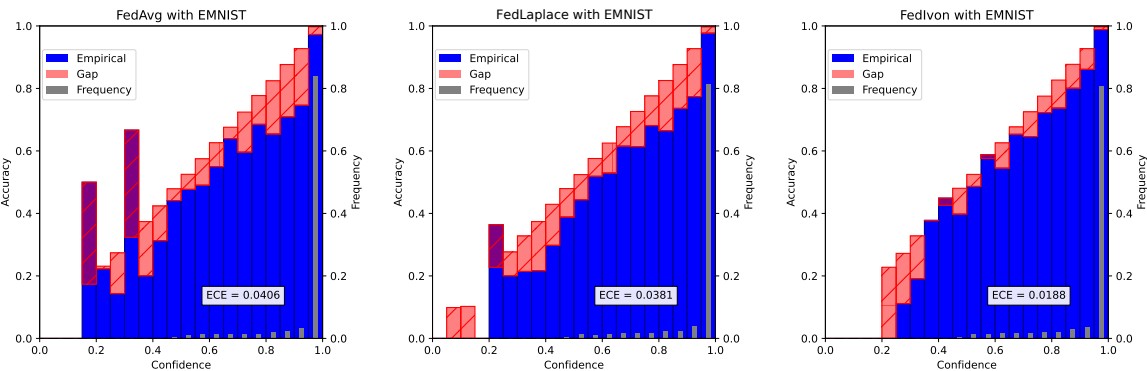

Figure 5: Reliability diagrams for EMNIST experiments (left: FedAvg, center: Fedlaplace, right: FedIvon).

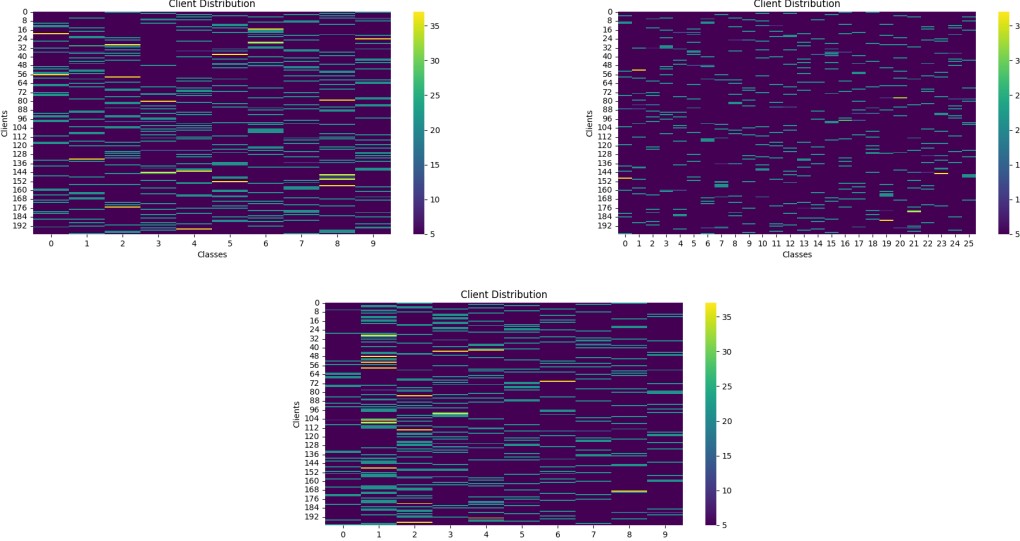

Figure 6: Client data distribution for CIFAR-10, EMNIST, and SVHN dataset used in FL experiments.

biased data from individual clients must be aggregated effectively to learn a model capable of generalizing across all classes. This scenario highlights the complexities and practical relevance of federated learning in real-world applications.

## G   Client data distribution in pFL

Figure 7 illustrates the distribution of data points across classes and clients in three pFL experimental setups with 50, 100, and 200 clients. The number of data points per client varies significantly, with some clients having over 1,000 data points and others fewer than 5, indicating a high degree of imbalance. Despite this, every client retains examples from most classes, which is crucial for training personalized models that adapt to the unique data distribution of each client. This setup highlights the challenge of learning effective personalized models in pFL. Similarly, Figure 8 shows the data distribution for the CIFAR-100 dataset.

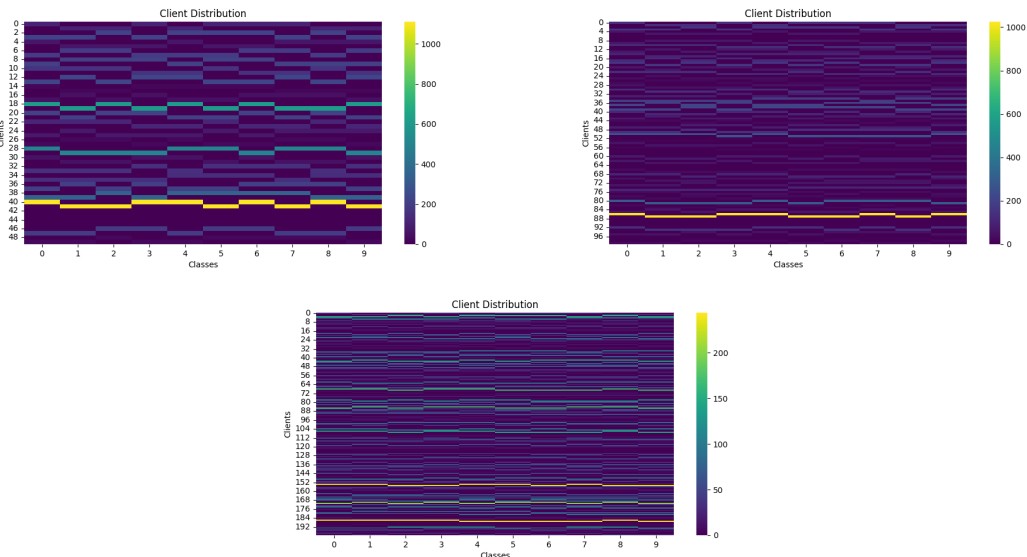

Figure 7: Client data distribution for CIFAR-10 dataset used in pFL experiments (left: 50 clients, right: 100 clients, bottom: 200 clients).

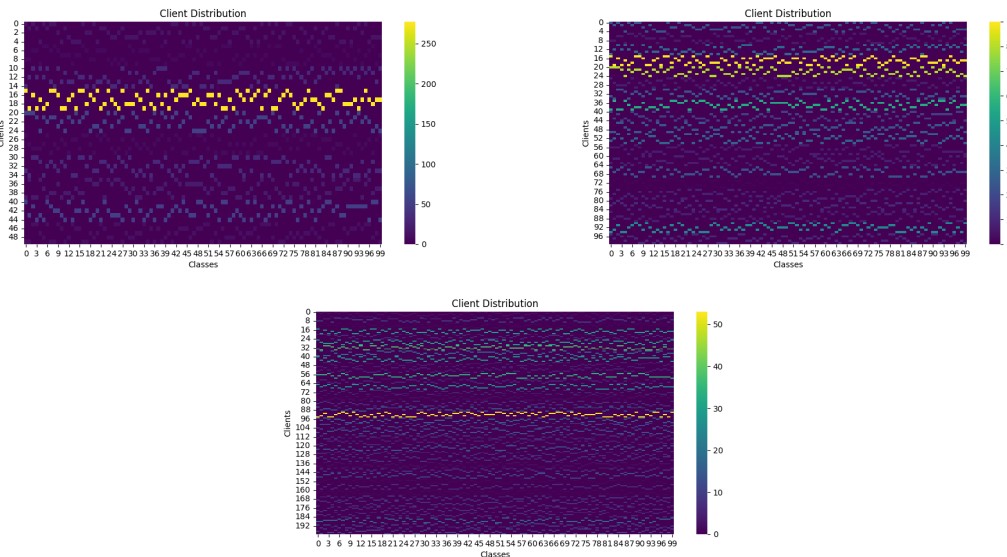

Figure 8: Client data distribution for CIFAR-100 dataset used in pFL experiments (left: 50 clients, right: 100 clients, bottom: 200 clients).

