# OpenReview forum: "Federated Learning with Uncertainty and Personalization via Efficient Second-order Optimization"
_TMLR — Accepted by TMLR_

### Review · Reviewer_Q1Kh · 2025-10-17

**Summary Of Contributions:**

This paper introduces FedIvon, a novel federated learning (FL) algorithm that brings Bayesian uncertainty estimation and personalization into FL while remaining highly efficient. The core idea is to perform variational Bayesian inference on each client using an Improved Variational Online Newton (IVON) optimizer, which is a second-order (Newton-like) method that approximates the Hessian implicitly. By leveraging IVON, each client maintains a Gaussian posterior with a diagonal covariance (uncertainty) over the model parameters, but the computational cost per communication round is kept comparable to standard first-order methods like Adam. This means FedIvon can capture curvature information for faster convergence and uncertainty quantification without the heavy overhead typically associated with second-order methods or exact Bayesian updates. Another key contribution is that FedIvon naturally supports personalized models for each client in a hierarchical Bayesian manner: the server aggregates client posteriors into a global posterior which then serves as a prior for subsequent client updates. This hierarchy allows each client to personalize its model to its local data while still benefiting from global knowledge sharing. In summary, the paper’s main technical contributions are:

- Efficient Bayesian Federated Optimization: An FL algorithm that uses second-order variational optimization (via IVON) to approximate each client’s posterior. It achieves per-round computation similar to Adam but yields better predictive performance and better-calibrated uncertainty estimates than both classic FL (point estimate) and prior Bayesian FL approaches. The method implicitly computes a diagonal Hessian approximation on-the-fly, avoiding expensive explicit Hessian calculations as required by methods like FedLaplace. This is a strength because it bridges the gap between Bayesian FL’s benefits and practical efficiency/scalability in resource-constrained settings.

- Uncertainty Quantification and Personalization: By maintaining a Gaussian posterior for each client model, FedIvon provides predictive uncertainty estimates (via the variances) in addition to point predictions. The uncertainty information is used for tasks like model calibration and out-of-distribution detection. Moreover, in a hierarchical Bayes fashion, the global model’s posterior (mean and covariance) acts as a prior for clients, enabling client-specific personalized models that account for local data heterogeneity. This personalization is seamlessly integrated into the federated updates (clients receive the server’s posterior and update their own posteriors accordingly), which is an elegant way to handle non-i.i.d. data.

Overall, the paper’s contributions tackle an important problem that is combining Bayesian FL with efficiency. A notable strength is the comprehensive experimentation demonstrating improved accuracy and calibration. One potential weakness in the contributions is that the approach builds upon an existing optimizer (IVON from Shen et al., 2024) and known Bayesian personalization frameworks, so the novelty is more in the integration and efficiency gains in FL context rather than an entirely new theory. Another limitation worth noting upfront is that FedIvon assumes a diagonal covariance for tractability. This is pragmatic, but it means uncertainty modeling is limited to diagonal (no correlations between parameters). This could restrict the expressiveness of the posterior, though it keeps computation feasible.

**Audience:**

Yes

**Audience Explanation:**

Yes. The topics and findings of this paper are likely to interest a segment of TMLR’s audience, particularly those working in federated learning, Bayesian deep learning, and uncertainty in machine learning. Federated learning is a well-established research area, and one of its current frontiers is how to handle heterogeneous data across clients (non-i.i.d.) and how to provide trustworthy models with uncertainty estimates. This paper squarely targets both issues by proposing a Bayesian personalization approach. The idea of using hierarchical Bayesian models in FL (i.e., a global prior and client-specific posteriors) has been gaining traction recently, and this work provides an efficient way to do it, which would be of interest to researchers who found previous Bayesian FL methods too cumbersome or impractical.

**Broader Impact Concerns:**

The paper’s broader impact discussion (if any) appears to be minimal, so here I raise a few concerns and considerations that should be acknowledged:

- Fairness Across Clients: Personalized federated learning can inadvertently introduce fairness concerns. Since each client learns its own model, differences in data quantity or quality across clients might lead to some clients obtaining much better models than others.

- Reliability of Uncertainty Estimates: While FedIvon improves calibration overall, incorrect uncertainty estimates can be harmful if relied upon in high-stakes settings. A concern is whether the model might be overconfident on out-of-distribution or minority group data that it did not see (especially if a client’s training data is limited or biased).

- Privacy and Security: Federated learning is often praised for on-device privacy, but sharing model updates (means and variances in this case) can still leak information about clients’ data. The paper does not discuss privacy-preserving enhancements (like differential privacy or secure aggregation). A potential impact risk is that malicious servers or other clients might infer personal information from the transmitted posterior parameters.

**Claims And Evidence:**

Yes

**Claims Explanation:**

Yes, for the most part the claims are supported by convincing evidence, through arguments and experiments.

The submission’s central claims are: (1) FedIvon achieves superior predictive performance and uncertainty calibration compared to state-of-the-art baselines, (2) it incurs minimal overhead (computation and communication) similar to first-order methods, and (3) it enables effective personalization for heterogeneous clients. The evidence provided to back these claims is generally solid, with a few gaps as discussed below:

- Technical Correctness and Clarity: The algorithm is described with sufficient clarity. The paper includes pseudocode for the client update (IVON) and the federated aggregation (Algorithm 1 and 2), which makes it easy to verify what exactly is implemented. The derivation of how client posteriors are combined into a global posterior is grounded in known results (product-of-Gaussians formula) and is clearly explained. The method for personalization is also described clearly. These descriptions suggest the approach is technically sound. Important details like how the Hessian is estimated on clients (via a reparametrization trick yielding an unbiased diagonal Hessian estimate) are given, showing that IVON indeed provides second-order information without high cost. The authors make a convincing argument that there is no explicit costly Hessian inversion or large matrix communication; instead, only diagonal variance vectors are maintained, which is efficient. This lends credibility to the claim of “efficient second-order optimization.” However, one concern is that the paper does not formally prove convergence or provide a theoretical complexity analysis as it relies on references and intuition (e.g. IVON’s known efficiency) to support these points. In the absence of theoretical proofs, the clarity of the algorithm and the empirical evidence become even more crucial, and fortunately the empirical results do back up the efficiency and performance claims.

- Empirical Support for Performance Claims: The experimental evaluation is thorough and directly addresses the claims:

  - *Accuracy and Calibration:* In standard federated learning experiments (classification on EMNIST, CIFAR-10, SVHN), FedIvon consistently outperforms the baselines in terms of test accuracy and yields much better calibrated predictions. Table 1 compares FedIvon against a representative strong FL baseline (FedAvg) and a recent Bayesian FL method (FedLaplace) on multiple metrics. FedIvon achieves the highest accuracy on all three datasets and dramatically lower calibration error (ECE) and negative log-likelihood, indicating its probabilistic outputs are more trustworthy. These results support the authors’ claim that FedIvon improves predictive performance and uncertainty quantification over both non-Bayesian and Bayesian competitors.

  - *Out-of-Distribution (OOD) Detection:* The authors also test whether the improved uncertainty translates to better OOD detection. Using the entropy of predictions to detect when an input is outside the training distribution, FedIvon achieved the highest or equally-high AUROC on most dataset pairs. In Table 2, FedIvon’s OOD detection AUROC is slightly better than or on par with FedAvg and clearly better than FedLaplace in two of three cases. (It’s worth noting one nuance: on CIFAR-10 as in-distribution, FedAvg had a slightly higher OOD AUROC than FedIvon (0.7896 vs 0.7662). The authors describe FedIvon’s OOD performance as “better or competitive,” which is fair. FedIvon was best on EMNIST and SVHN scenarios, and a close second on CIFAR-10. This small discrepancy doesn’t undermine the overall claim, but it suggests FedIvon’s uncertainty is at least not worse than FedAvg’s in any case, and usually better.)

  - *Personalization (Heterogeneous data):* A significant claim is that FedIvon can personalize effectively for each client thanks to the hierarchical Bayesian approach. The evidence for this comes from experiments simulating non-i.i.d. scenarios on CIFAR-10 and CIFAR-100 (with different types of data heterogeneity). The paper compares FedIvon to strong personalized FL baselines including pFedME (a state-of-the-art optimization-based pFL method), pFedBayes (a recent Bayesian personalized FL), and pFedVEM (variational expectation maximization approach), across varying numbers of clients. The results show that FedIvon achieves equal or higher personal-model accuracy than these methods and significantly higher global-model accuracy.

  - *Efficiency and Scalability:* The paper’s efficiency claims are mostly supported qualitatively and through algorithmic analysis rather than direct runtime plots. The authors argue that per-round computation of FedIvon “matches first-order optimizers (e.g., Adam)” and that it does not incur the usual overheads of second-order methods in FL. This is plausible given that IVON’s updates require similar operations to Adam (plus simple variance updates) and avoid expensive steps like Hessian inversion or MCMC sampling. While the experiments report using up to 2000 communication rounds and reasonably large numbers of clients (up to 200) without issues, the paper does not explicitly graph runtime or communication costs. The submission could be stronger by quantitatively showing the runtime per round or total training time for each method, but the qualitative description and the known properties of IVON give confidence that the method is indeed efficient. Moreover, the authors explicitly discuss why FedIvon’s initial convergence can be slightly slower (due to stochastic weight sampling for gradients) and show an ablation varying local training epochs; this kind of analysis indicates they have a good understanding of the method’s behavior and that any efficiency trade-offs (like a slower start) are minor and well-managed.

  - *Adequacy of Baselines:* The baselines chosen are mostly appropriate and help validate the claims, though there is room for a couple more comparisons. For standard FL (non-personalized), the paper benchmarks against FedAvg (the classic method) and FedLaplace. FedAvg is a natural baseline for accuracy (point estimate, no uncertainty), and FedLaplace (Liu et al., 2024) is a recent Bayesian FL method that also approximates client posteriors as Gaussians via Laplace’s method. FedLaplace is a strong baseline because it represents the idea of Bayesian FL with second-order info (though it computes Hessians explicitly at the clients). The results showed FedIvon outperforming both, which supports the claim of being state-of-the-art. One might question if other baselines like FedProx (for non-iid data) or FedPAshould have been included. Overall, the evidence provided is sufficient and the baselines are adequate to support the main claims. Only minor additions (like one more baseline in the global setting, or perhaps comparing to an ensemble approach such as FedBE) could further strengthen the empirical support, but their absence does not invalidate the claims.

**Requested Changes:**

- Include Additional Baselines or Justify Their Exclusion: To solidify the SOTA claims, the authors should include at least one additional baseline in standard FL experiments: e.g., FedPA (posterior averaging), FedBE (Bayesian ensemble distillation), or FedProx (a widely used FL method for non-i.i.d. data). If these methods are omitted, please explicitly justify. A brief discussion of how FedIvon differs from and improves upon FedPA in particular would clarify novelty.

- Add Theoretical Insight or Stability Guarantees: The paper currently lacks theoretical support. Please either (a) sketch convergence behavior using results from the IVON literature, or (b) include a qualitative explanation of why FedIvon remains stable in the federated setting (e.g., guaranteed positive diagonal Hessian, bounded variance updates, etc.).

- Clarify Communication and Computational Overhead: Please describe the communication cost per client per round (e.g., sending mean and variance vectors = 2× model size) and clarify why this remains practical. Also discuss computational complexity—IVON appears to require two forward-backward passes per batch, similar to Adam. If available, please provide runtime estimates or FLOP comparisons to support the claim of efficiency.

- Discuss Limitations of the Method: A short limitations paragraph would improve balance. Points to mention include: the diagonal covariance assumption (no parameter correlation), added memory cost from tracking variances, client-side compute overhead, and the fact that experiments were conducted on relatively small CNNs in the vision domain. Acknowledging these would help set realistic expectations.

---

### Review · Reviewer_Kt8z · 2025-11-04

**Summary Of Contributions:**

This work proposed a novel Bayesian federated learning (FL) method, where each client performs Ivon and updates its local weight and Hessian, and the server updates the weight and Hessian with weighted update of each client. The method is significant as a second-order optimization with efficiency of first-order method. Besides, the FedIvon framework also enables client personalization. The significance of the method is supported by empirical results on various datasets.

Strength:
- The paper is well structured and well written. The background is clearly introduced, the challenges and state-of-the-art methods are summarized.
- The paper adapts the Ivon algorithm and deploys it in FL setting.
- The empirical results are strong, the FedIvon approach achieves better metrics in data heterogeneous settings compared with the baselines.

Weaknesses:
- The paper does not invent Ivon algorithm, therefore partially limiting its originality.
- Since the local update subsumes Ivon algorithm, whichever limitations Ivon has will also be inherited in the FedIvon framework.
- More baselines of Bayesian FL are mentioned in related work but not compared with in experiments.
- To show the efficiency, maybe experiments on runtime can be added, e.g., compare with FedAvg and Fedlaplace like Figure 4, but the x-axis being runtime.

**Audience:**

Yes

**Audience Explanation:**

This work is novel and enough contribution to FL community.

**Claims And Evidence:**

Yes

**Claims Explanation:**

The FedIvon framework compares with FedAvg and FedLaplace in terms of Bayesian FL, and clearly exhibits better results on EMNIST, SVHN and CIFAR datasets in terms of convergence, accuracy, loss, and uncertainty. Besides, FedIvon is compared with pFedME and pFedBayes baselines, and shows better results on CIFAR datasets.

**Requested Changes:**

- While this work shows significant improvement with Ivon algorithm on clients, limitation of the FedIvon framework should be more discussed.
- Maybe add more comparisons with Bayesian FL methods.
- Maybe add experiments showcasing the efficiency and computational overhead of FedIvon compared with FedAvg and FedLaplace.

---

### Review · Reviewer_UPt5 · 2025-11-08

**Summary Of Contributions:**

### Summary of Contributions

The paper addresses the challenges of federated learning (FL), where data cannot be shared across clients and communication is limited. Given $K$ clients with private datasets $\mathcal{D}_k$, the global goal in standard FL is to solve:

$$
\theta^* = \arg \min_\theta \sum_{k=1}^{K} -\log p(\mathcal{D}_k \mid \theta).
$$

However, directly solving this is difficult due to data localization, and point-estimate solutions like FedAvg do not quantify uncertainty. Computing the full posterior $p(\theta \mid \mathcal{D})$ would be more informative as it captures model uncertainty, but exact Bayesian inference is computationally intractable, and common approximate methods (e.g., MCMC, Laplace approximation, VI) are too expensive for FL.

To overcome these limitations, this paper proposes **FedIvon**, a Bayesian FL framework that uses **Improved Variational Online Newton (IVON)** to efficiently estimate a posterior distribution over model parameters at each client. Instead of learning a single global model, FedIvon learns client-wise posterior distributions characterized by a mean and diagonal covariance. This enables:
- principled **uncertainty quantification**, and
- **natural personalization** via hierarchical Bayesian modeling, where each client adapts from a shared prior.

The key contribution is demonstrating that Bayesian FL can be performed with **near-first-order computational and communication cost**, avoiding heavy matrix operations or full posterior transmissions. FedIvon brings uncertainty estimation and personalization without the typical overhead of MCMC or Laplace-based approaches, while improving accuracy and calibration over existing FL baselines.

**Additional Comments:**

My recommendation is to accept after minor revisions. The work is not highly novel from a theoretical standpoint, but the empirical results and integration are solid and align with TMLR’s standards, which emphasize usefulness and clarity over pure novelty.

**Audience:**

Yes

**Audience Explanation:**

The paper addresses Bayesian federated learning, uncertainty quantification, and personalized FL; topics that are highly relevant to the TMLR community. Researchers working on federated learning, Bayesian deep learning, or efficient optimization methods would likely find these contributions valuable.

**Broader Impact Concerns:**

I do not see any ethical concerns.

**Claims And Evidence:**

Yes

**Claims Explanation:**

The paper provides strong empirical support for its claims regarding predictive accuracy, calibration, and uncertainty estimation. The experiments are well-designed, cover multiple datasets, and include both standard and personalized FL settings.

However, a key contribution of the paper is the claim that FedIvon achieves computational efficiency comparable to first-order optimizers while being more efficient than other Bayesian FL methods. This is only argued qualitatively. I could not find quantitative evidence such as runtime, FLOPs, memory usage, or per-round communication cost. As a result, while the empirical results convincingly support the performance and uncertainty-related claims, the efficiency-related claims are not fully substantiated by measurable evidence.

**Requested Changes:**

$\mathbf{1.}$ The paper claims that FedIvon has computational cost comparable to first-order optimizers like Adam and is more efficient than existing Bayesian FL methods. However, this argument is presented qualitatively. I was unable to find quantitative evidence such as runtime, FLOPs, memory usage, or communication cost comparisons. Providing such empirical results would make this contribution more convincing and further strengthen the paper.

$\mathbf{2.}$ Clarify the limitations of the diagonal covariance assumption.
The method approximates posterior distributions using diagonal Gaussians. A brief discussion on when this assumption may break down or how it might affect uncertainty calibration would improve the transparency and rigor of the work.

$\mathbf{3.}$ Discuss the choice of only using 2 local epochs.
In most federated learning setups, the number of local training epochs is typically greater than 2 (e.g., 5 or 10), especially when communication is costly. It would strengthen the paper to either justify this choice or include results for higher values of local epochs to better understand convergence, stability, and communication–computation trade-offs.

---

### Decision · Action_Editor_jAZi · 2025-12-19

**Recommendation:** Accept with minor revision

**Additional Comments:**

The paper presents a Bayesian federated learning framework that maintains per-client uncertainty summaries (via a diagonal Gaussian approximation) and aggregates them on the server to support both uncertainty quantification and hierarchical personalization across rounds. A central ingredient is the IVON optimizer (Shen et al., 2024), which the authors adapt to the federated setting to obtain a practical second-order variational update with a near-first-order cost.

The reviewers were broadly aligned in their assessment that the work is technically sound, clearly written, and sufficiently supported by experiments. The main reservations were about the level of conceptual novelty, given the reliance on an existing optimizer and established Bayesian FL ingredients, as well as the need for more explicit evidence and discussion regarding efficiency and limitations. The revision responds well to these points, particularly by incorporating quantitative efficiency measurements and expanding the discussion of limitations.


### Minor Revision:
Please carefully review the final version for minor typos and typesetting issues. Eg.
- on page 1: "FedAvg(McMahan" -> "FedAvg (McMahan";
- Figure 3 & 5: "FedLapalce" -> "FedLaplace"
- on page 11: "work Zhu et al. (2023)" -> "work (Zhu et al., 2023)".

**Audience:**

Yes

**Audience Explanation:**

The topic is within TMLR’s scope. The submission substantially builds upon the IVON optimizer introduced by Shen et al. (ICML 2024) and adapts it to a federated Bayesian setting, with an emphasis on uncertainty estimation and personalization. The resulting approach should be of interest to readers working on federated optimization, uncertainty in deep learning, and Bayesian or variational methods

**Claims And Evidence:**

Yes

**Claims Explanation:**

The manuscript presents a well-specified method and supports its main claims with an experimental evaluation covering both standard federated learning and personalized settings. The paper does not aim to establish new theoretical guarantees, nor does it provide formal convergence proofs. The claims are verified through an empirical evaluation, complemented by a qualitative discussion on stability.

A key concern raised in the initial reviews was the lack of quantitative evidence to support the efficiency claims. The revised version addresses this directly by adding concrete measurements of per-round computation and communication costs, making the tradeoffs explicit and verifiable for the problem scales evaluated in the submission.